

# Network models of driver behavior

Markus T. Mattsson[1,2]

[1] Department of Psychology and Logopedics, University of Helsinki, Helsinki, Finland
[2] Traffic Research Unit, University of Helsinki, Helsinki, Finland

## ABSTRACT

The way people behave in traffic is not always optimal from the road safety perspective: drivers exceed speed limits, misjudge speeds or distances, tailgate other road users or fail to perceive them. Such behaviors are commonly investigated using self-report-based latent variable models, and conceptualized as reflections of violation- and error-proneness. However, attributing dangerous behavior to stable properties of individuals may not be the optimal way of improving traffic safety, whereas investigating direct relationships between traffic behaviors offers a fruitful way forward. Network models of driver behavior and background factors influencing behavior were constructed using a large UK sample of novice drivers. The models show how individual violations, such as speeding, are related to and may contribute to individual errors such as tailgating and braking to avoid an accident. In addition, a network model of the background factors and driver behaviors was constructed. Finally, a model predicting crashes based on prior behavior was built and tested in separate datasets. This contribution helps to bridge a gap between experimental/theoretical studies and self-report-based studies in traffic research: the former have recognized the importance of focusing on relationships between individual driver behaviors, while network analysis offers a way to do so for self-report studies.

## INTRODUCTION

It has become something of a truism in human factors research that people contribute to road crashes by either deliberately violating rules or by making unintended errors. In traffic psychology, perceiving the importance of this distinction coincides with the development of a much-used questionnaire instrument, the driver behavior questionnaire (DBQ) (*Reason et al., 1990*). A seminal study (*Reason et al., 1990*) hypothesized that the distinction might be due to violations and errors being "mediated by different psychological mechanisms," and interpreted the results of a principal components analysis (PCA) from this point of view. The idea that there exist fundamentally different types of "aberrant behavior" that need to be targeted by different types of intervention has since appeared in numerous research articles, some of which have been based on the errors/violations dichotomy, some of which have made finer distinctions between different types of errors and violations. Subsequent studies have commonly referred to *Reason et al. (1990)* as having proved the existence of

Corresponding author
Markus T. Mattsson,
markus.mattsson@helsinki.fi

two qualitatively different psychological mechanisms even though the authors of the original study were careful to note that the distinction was hypothetical in nature. The idea is echoed in statements such as: "Errors and violations result from different psychological processes (*Reason et al., 1990*) and therefore should be treated differently" (*Mesken, Lajunen & Summala, 2002*) "Since errors and violations result from different psychological processes, they should be treated differently," (*Reason et al., 1990*; *Lajunen, Parker & Summala, 2004*) and "As each type of behavior has a distinct psychological underpinning (*Reason et al., 1990*), different interventions are required to reduce their frequency and also associated crash risk" (*Stephens & Fitzharris, 2016*). This idea is herein referred to as the *latent variable view of violations and errors.*

The received wisdom of the latent variable view has gone unchallenged for two main reasons. First, a sufficiently detailed theory on the relationships between psychological mechanisms, traffic behaviors and latent variables is still lacking. Second, individual violations and errors may well be causally related: for example, exceeding the speed limit (a violation) may cause one to miss observing something (an error). The present paper focuses on the latter issue, and *conceptualizes driver behavior in a novel manner as an interacting network of behaviors, emotional reactions and perceptions.* Similar network models have been recently employed in research on psychopathology (*Borsboom & Cramer, 2013*; *Fried & Cramer, 2017*), personality (*Costantini et al., 2015*), attitudes (*Dalege et al., 2016*) and intelligence (*Van Der Maas et al., 2006*) as descriptions of how the phenomena under investigation may arise from the interactions of their component parts.

## Psychological mechanisms and latent variables in traffic psychology

Consider the above-cited assertion that dimensionality reduction produces insights about psychological mechanisms. This is unlikely, since a mechanistic explanation of a cognitive process involves accounting for the component operations of the process and their interaction (*Lappi & Rusanen, 2011*; *Bechtel, 2008*), while PCA and factor analysis are—stated simply—procedures for grouping correlated variables. In fact, offering violation- and error-proneness as explanations for why individuals break rules and commit errors amounts to reified circular reasoning (*Boag, 2011*) insofar as the "pronenesses" refer to nothing but being likely to behave in said manner. Furthermore, it has been argued that a detailed analysis of psychological mechanisms should precede the generation of psychometric instruments instead of being used as a post hoc explanation for the correlational structure of the instrument in the data analysis stage (*Embretson, 1994*). When latent variable models have been constructed in this manner, based on theories of cognitive psychology, it has been observed that several psychological mechanisms may contribute to a given latent variable score (*Borsboom, Mellenbergh & Van Heerden, 2003*; *Embretson, 1994*).

Further, the individual DBQ errors and violations are of such a wide variety that it is unclear how they could all be implemented by a single cognitive or psychological mechanism. For instance, the errors include missing an observation, forgetting the road one was travelling, ending up on an often-used route when intending to travel elsewhere, etc. Accordingly, it has been argued that these errors are differentially related to

top-down and bottom-up attention, updating of a mental model of the driving situation and resolving conflicts between schemas in memory (*Mattsson, 2012*). Similar considerations apply to individual violations. Network models, with their focus on the interactions between traffic behaviors, may prove useful in practice in encouraging the researcher to focus on the psychological underpinnings of individual traffic behaviors.

## Causal and statistical assumptions of latent variable models and network models

When discussing latent variable models, a distinction is often drawn between measurement models and structural models (*Kline, 2011*). Measurement models describe the relationships between latent variables (e.g., errors) and observed variables (e.g., miss observing a pedestrian, misestimate the speed of another vehicle). Structural models concern interrelationships between latent variables (and observed variables assumed to be measured without error). Now, as the name suggests, in a measurement model, the relationship between the latent and observed variables is understood as one of measurement. This idea is most readily compatible with giving the latent variables a realist interpretation, that is, understanding them as existing independently of the act of measurement (*Borsboom, Mellenbergh & Van Heerden, 2003*). Even though the matter is rarely explicitly discussed in traffic psychology, the realist interpretation seems in line with how latent variables are used in practice: separate psychological processes are taken to underlie the latent variables (errors and violations), and causal language is used when describing the relationships between the latent and observed variables (*Mesken, Lajunen & Summala, 2002*; *Lajunen, Parker & Summala, 2004*; *Stephens & Fitzharris, 2016*). Under the realist interpretation, the relationship between the latent variable (the measured quantity) and observed variables is seen as one of causation: the measured quantity is seen as causing variation in the observed variables similarly to how temperature causes variation in thermometers (*Schmittmann et al., 2013*). It is naturally possible to interpret latent variables in a non-realist fashion, for example, as mere statistical summaries, in which case the relationship can be understood as a logical one instead of a causal one (*Borsboom, Mellenbergh & Van Heerden, 2003*). However, if one is committed to ideas such as estimating the values of parameters, evaluating the position of a subject on the latent variable or assessing the truth of a claim involving a latent variable, it appears that latent variables need to be given a realist interpretation (*Borsboom, Mellenbergh & Van Heerden, 2003*).

The realist interpretation of latent variables is naturally compatible with viewing variation in the observed variables as being due to either the latent variables or measurement error, and allowing no covariance among the observed variables once these effects are accounted for *Borsboom, Mellenbergh & Van Heerden (2003)*. Such relationships are schematically depicted in Fig. 1. Attributing causal power to latent variables in this manner renders the observed variables inert, interchangeable reflections of the latent properties being measured (*Borsboom, Mellenbergh & Van Heerden, 2003*; *Schmittmann et al., 2013*). The idea that observed variables are rendered statistically
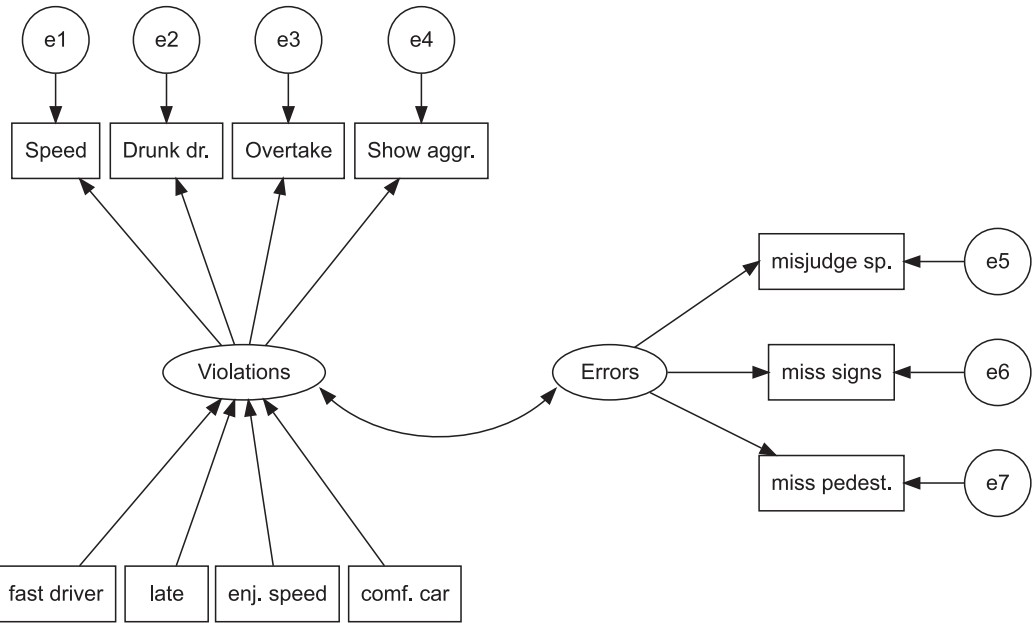

**Figure 1 The latent variable view of violations and errors.** Rectangles refer to observed variables, circles and ovals to latent variables. The ovals with the letter "e" and a number refer to error variances of the respective observed variables. The error variances (e1–e7) are assumed uncorrelated as per the assumption of conditional independence. Arrows pointing in one direction refer to assumed causal associations, while the two-headed arrow represents a covariance relation. To simplify the figure, predictors for errors are not shown.

independent when conditioning on the latent variable(s) is known as the assumption of local independence (*Borsboom, Mellenbergh & Van Heerden, 2003*) and it is problematic in at least two cases: (1) when several latent variables prove to influence a given set of observed variables (lack of unidimensionality) and (2) when direct causal connections exist among the observed variables.

The problematic nature of the unidimensionality assumption is concretely reflected in the common practice of representing errors and violations as unweighted or weighted (*De Winter, Dodou & Stanton, 2015*) sum scores. The following example is related to the unidimensionality (or the lack thereof) of items related to the subscale of violations in the DBQ. Consider two imaginary persons filling in the DBQ: John, known for his quick temper, answers the three items related to aggressive behavior with the option "nearly all the time," and reports performing no other violations, thus obtaining the sum score of 21. Bill, on the other hand, known for his careful nature, chooses the option "never" to the aggression-related items and the option "hardly ever" or "occasionally" to the other violation items. As there are many more items related to non-aggressive violations than to aggressive ones, both respondents receive identical scores, even though their behavioral profiles are quite different. It is naturally possible to react to this challenge within the latent variable framework by creating a more complex model. For instance, "ordinary violations" and "aggressive violations" can be modelled as separate latent variables (*Mattsson, 2012*). Still, the unidimensionality of the items related to these further latent variables remains an open research question.

The latter idea, the possibility of direct causal connections among observed variables is less often discussed, and it motivates applying network models in the first place. For instance, driving under the influence of alcohol or driving fast may predispose the driver to other violations and errors, and the consequences of these two violations are likely to be different. Assuming the existence of such direct associations between observed variables is in fact quite a drastic move, since it necessitates viewing the relationship between the latent and observed variables as something else than measurement, perhaps as one of constitution (*Cramer et al., 2012b*). The metaphor of interacting birds forming a flock has been used to describe the nature of the relationship (*Cramer et al., 2012b*).

Finally, previous research has indicated equivocal factorial structures for the DBQ. This is evidenced by

1. Different factor structures being obtained for the same version of the instrument. For instance, the often-used 27-item questionnaire is thought to reflect two (*De Winter, 2013*; *De Winter & Dodou, 2010*), three (*Parker et al., 1995*) or four (*Rowe et al., 2015*) different psychological processes, with exploratory analyses sometimes indicating even more latent variables (*Stanojević et al., 2018*);
2. Typically high cross-loadings of items on factors (*Mattsson, 2012*; *Mattsson et al., 2015*);
3. Complex factor structures needed to adequately fit the data, either by specifying second-order factors (*Lajunen, Parker & Summala, 2004*) or a general factor (*Stanojević et al., 2018*) and
4. Failures of the test of measurement equivalence across certain subgroups (*Stephens & Fitzharris, 2016*; *Mattsson, 2012*; *Mattsson et al., 2015*).

Observations such as these function as an empirical motivation for creating the network models reported in the present paper.

Figure 1 shows schematically the fundamental assumptions of latent variable models. Individual behaviors, such as speeding or misjudging speed, are assumed to reflect the level of the underlying latent variable and measurement error. Background factors such as enjoying speed figure as predictors of the latent variables. Importantly, background factors are unrelated to individual driver behaviors, which are assumed causally inefficacious. Typically, a dependent variable of interest, such as the number of crashes, is regressed on sum variables representing violations and errors (*De Winter, Dodou & Stanton, 2015*). One distinct benefit of such models is their simplicity: if the latent variables indeed manage to capture the important commonalities among the individual behaviors, they provide an extremely parsimonious representation of the data. In addition, the model shown in Fig. 1 is, on purpose, a rather simple example of a structural equation model; these models are a flexible tool that would allow the researcher to add more latent variables and to represent much more complex relationships among them. Further, the model shown in Fig. 1 is not intended to summarize theoretically or practically important findings in traffic research, but rather, together with Fig. 2, to illustrate the kinds of relationships that may obtain between the different types of variables in latent variable models and network models, respectively.

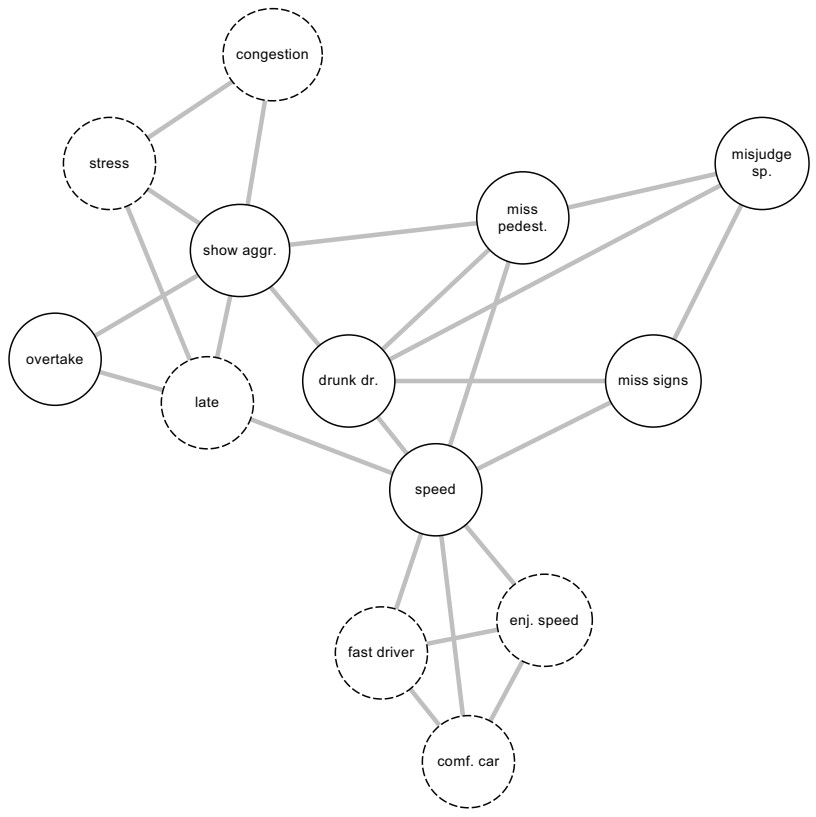

**Figure 2 The network view of traffic behavior.** Traffic behaviors encoded in the DBQ are drawn using solid lines, background factors using dashed lines. Pairwise relationship among individual violations and errors, when controlling for the effects of all other variables, are shown as the edges of the network. The background variables "congestion" and "stress" that were not included in Fig. 1 have been added to indicate the possibility that there exist background variables that are related to only one of the driver behaviors included in the model.

Network models, such as the hypothetical model shown in Fig. 2, are naturally suited to investigating direct associations among traffic behaviors. Individual behaviors are also assumed to be directly affected by background factors. Further, two background variables, stress and congestion, that were not included in Fig. 1, are shown in Fig. 2. This illustrates the possibility that there may exist background factors that are related to only one of the observed variables. Such relationships can be more naturally accommodated in network models than in latent variable models. In network models, the absence of an edge between two nodes is interpreted as showing that when the effects of all other nodes in the network are controlled, the two nodes are rendered statistically independent (*Epskamp & Fried, 2018*). The non-zero edges can be interpreted as potential causal connections (*Epskamp & Fried, 2018*), logical relationships of entailment (*Kossakowski et al., 2016*), reflections of the influence of an unmodeled latent variable affecting both nodes (*Epskamp, Kruis & Marsman, 2017*), conditioning on a collider (a common effect of both nodes) (*Epskamp et al., 2018*) or perhaps semantic relationships of the nodes being close to each other in meaning. Finally, a cluster of nodes may reflect the effect of a latent variable influencing all the nodes in the cluster (*Golino & Epskamp, 2017*).

Network models are valuable tools in exploratory analysis of the data: they enable representing pairwise associations among variables quickly, intuitively and efficiently. However, network models assume that the relationships among the observed variables (in this case, the driver behaviors) are not due to latent variables at all, which is a remarkably strong assumption. In this sense, latent variable models and network models can be seen as having complementary strengths and weaknesses (*Epskamp, Rhemtulla & Borsboom, 2017*).

This paper conceptualizes the behavior of newly licensed drivers as an interacting network of component behaviors. First, the dynamics of the relationships between the typical levels of individual violations and errors during the first 3 years post-licensure are represented as the between-person network (*Costantini et al., 2019*) which enables assessing direct associations between behaviors and judging the centrality of the behaviors in the network. Second, relationships between driver behaviors and background factors are assessed in a cross-sectional analysis using data collected at 6 months post-licensure. Importantly, the background factors are directly linked to individual driver behaviors rather than latent variables or sum scores. Third, individual driver behaviors recorded at 6 months post-licensure are used for predicting crashes occurring during the following 2 ½ years. Different predictive models are compared by building the models in one subset of data and testing them in another one.

# MATERIALS AND METHODS

## Data and participants

This study is based on the archival dataset collected in the longitudinal Cohort II study in 2001–2005 on new and novice drivers in the UK (*Wells et al., 2008*). The initial sample size was 20,512 with four waves of data collection taking place after the respondents had passed their driving tests. The numbers of responses and response rates were as follows: 10,064 at 6 months (49%), 7,450 at 12 months (36%), 4,189 at 24 months (26%) and 2,765 at 36 months (26%). The age distribution of the respondents corresponded to that of the population of newly licensed drivers, with 59% of respondents under the age of 20 at the first wave of data collection and 76% under the age of 25. Women, on the other hand, were overrepresented: at the first wave, 64% of the respondents were females. The dataset has been previously used in several studies (*De Winter, Dodou & Stanton, 2015*; *De Winter & Dodou, 2010*; *Rowe et al., 2015*; *Roman et al., 2015*) and is freely available online (*Transport Research Laboratory, 2015*).

The cross-sectional network analyses were performed on data obtained at 6 months post-licensure. Only cases with non-missing data in all variables were included, resulting in a sample size of 8,858. The respondents included in this analysis had a mean age of 22.51 years (SD = 7.95), and 64% of them were female. Due to the drop-out described above, a much smaller number of cases was available for forming the between-person network: the model was built based on 1,173 cases with no missing data. The between-person network was formed by averaging responses across the four time points. The respondents in the between-person analysis had a mean age of 24.04 years (SD = 9.62), and 71% of them were female. The regression analyses were based on cases

with no missing values on any of the predictor variables or the number of crashes variable, resulting in $N$ = 1,152, 69% female.

## Measures

Data collected using the Driving Experience Questionnaire (DEQ) that includes a 39-item version of the DBQ was used. In addition to the DBQ items, items referred to as "background factors" were used. The responses to the DBQ items were recorded on a six-point Likert scale ("never" to "nearly all the time"), and were intended to assess the two constructs of *violations* and *errors*. Example items include *How often do you disregard the speed limit on a motorway* and *How often do you select the wrong gear when wanting to go into reverse*, respectively. The response options for the background factors were as follows. Drivers' self-image was assessed on a seven-point scale anchored to the end points of a continuum (e.g., "inattentive" to "attentive"). The self-perceived improvement needs were recorded on a three-point scale (e.g., "no improvement needed in controlling the car" to "a lot of improvement needed in controlling the car"). Attitudes were assessed on a five-point scale ranging from "Strongly disagree" to "Strongly agree;" an example item reads *Decreasing the motorway speed limit is a good idea*. In addition, self-reported age, sex and mileage at 6 months post-licensure were used as predictors of public road crashes (both major and minor crashes) in the Poisson regression analyses.

## Procedure

The "Learning to drive questionnaire," which includes the question items related to attitudes, was filled in after the practical driving test, prior to responding to the DEQ. Informed consent was inferred from returned postal questionnaires in accordance with the social research guidelines of Department of Transport.

## Statistical analyses

### Network analyses

The network analyses were based on the common, widely used 27-item version of the DBQ (*Lajunen, Parker & Summala, 2004*; *Mattsson, 2012*, *2014*) together with items related to driving under the influence of alcohol and drugs, using the cell phone while driving and having to brake or swerve to avoid an accident, resulting in 31 DBQ items. These latter items are often omitted from latent variable models of the DBQ because of their low factor loadings (*Mattsson, 2012*), but they were now included due to their potential significance as determinants of other driving behaviors. Two types of network models were estimated. The cross-sectional network model was based on the 31 DBQ items together with nine items related to the background factors. The latter were chosen from among candidate variables using a series of exploratory network analyses (Figs. S1–S3). The between-person network (*Costantini et al., 2019*; *Epskamp et al., 2018*) was formed by calculating average scores for each respondent across the four time points to represent the overall pattern of driving behaviors during the first years of learning to drive. The benefit of between-person networks is that they are less susceptible than cross-sectional networks to spurious effects and reporting biases (*Costantini et al., 2019*)

such as mood-congruent recall (*Shiffman, Stone & Hufford, 2008*) as different biasing effects are likely to cancel each other out.

In both models, individual questionnaire items correspond to the nodes of the network, while the edges represent partial correlations controlling for all other nodes. The procedure attempts to uncover a graph known as a Gaussian Graphical Model in which each node is independent of the rest given the values of immediately neighboring nodes as described by the Markov properties (*Lauritzen, 1996*). In other words, an edge connecting two nodes indicates their conditional dependence given the other nodes. The edge weights were scaled to the joint maximum value of the two models to ensure the comparability of the results. All network analyses were performed in *R Development Core Team (2017)* using the packages qgraph (*Epskamp et al., 2012*) and bootnet (*Epskamp, Borsboom & Fried, 2017*).

Some of the partial correlations are likely to differ from zero because of sampling variation and can be thought of as false positive findings (*Costantini et al., 2015*). For this reason, the graphical lasso (*Friedman, Hastie & Tibshirani, 2008*) was used in estimating the networks based on polychoric correlations. The procedure constrains low values of partial correlations to zero, thus resulting in sparse models (*Epskamp & Fried, 2018*). The level of sparsity is determined by the tuning parameter $\lambda$, the value of which was chosen based on the extended bayesian information criterion (EBIC), as this has been shown to work well in retrieving the true network structure (*Foygel & Drton, 2010*) especially if the true model is sparse (*Epskamp & Fried, 2018*). The hyperparameter $\gamma$ used in EBIC model selection was set to the recommended default value of 0.5 (*Foygel & Drton, 2010*). The locations of the nodes were determined using a modified version of the Fruchterman–Reingold algorithm (*Fruchterman & Reingold, 1991*) for weighted networks (*Epskamp et al., 2012*), which places strongly connected nodes that have many edges in common close to one another.

The importance of the individual nodes in the network was assessed by calculating three indices of centrality: strength, betweenness and closeness. *Strength* refers to the sum of edge weights of the focal node, *closeness* to the reciprocal of the sum of distances from the focal node to all other nodes and *betweenness* to the number of shortest paths between two nodes that pass through the focal node. The values of the centrality indices were standardized to ensure comparability between networks and between studies. Further, the generalization of Zhang's local clustering coefficient (*Zhang & Horvath, 2005*) to signed networks (*Costantini & Perugini, 2014*) was calculated to represent the redundancy of the focal node. This coefficient was used due to its sensitivity to weak edges (*Saramäki et al., 2007*). The stability of the centrality indices and the accuracy of edge weight estimates were assessed by bootstrap analyses (Figs. S4–S5) using the bootnet package (*Epskamp, Borsboom & Fried, 2017*). Further, the stability of the indices was quantified by the correlation stability (CS) coefficient, the value of which should preferentially exceed 0.5 (*Epskamp, Borsboom & Fried, 2017*) (the values are reported in Table S5).

### Regression analyses

The 39 DBQ variables, together with the variables mileage, age and sex, recorded at 6 months post-licensure, were used as predictors of the number of crashes during the latter

waves of data collection in a Poisson regression model with a logarithmic link function. The regression analyses were based on all 39 DBQ items to maximize the predictive power of the model. The analysis was based on the idea of maximizing predictive accuracy via minimizing generalized cross-validation error (GCVE, operationalized as the deviation score), as described by *Chapman, Weiss & Duberstein (2016)*. The minimization of the GCVE aims at creating models that explain the maximum amount of variance without overfitting the model to data at hand. This is achieved by trading some increase in bias to a reduction in variance (*Chapman, Weiss & Duberstein, 2016*).

The data set was randomly split into a training set and a test set (75/25 ratio, with $N_{\text{training}} = 864$ and $N_{\text{testing}} = 288$). The uneven ratio was chosen to enable a sufficiently large number of cross-validation splits, with initial model fitting and cross-validation taking place within the training set, followed by fitting the same model in the test set using the R-package glmnet (*Friedman, Hastie & Tibshirani, 2010*). The same penalty, controlled by the regularization hyperparameter $\lambda$, was applied to all the predictors, which were standardized prior to analysis. Self-reported mileage was also log-transformed.

Three Poisson regression models were fit to the training and test data. First, an elastic net model (*Zou & Hastie, 2005*), was formed based on a grid search of optimal values of the hyperparameters $\alpha$ (elastic net mixing parameter) and $\lambda$ (regularization parameter), the values of which were chosen by minimizing the value of GCVE in a 10-fold cross-validation analysis. The elastic net combines penalties based on squared sums of regression coefficients (ridge) and the sum of their absolute values (lasso); it thus performs variable selection similarly to the lasso, and performs well with correlated predictors similarly to ridge regression (*Zou & Hastie, 2005*). Second, a ridge regression model (regularization without variable selection) was obtained by taking the cross-validated value of $\lambda$ (regularization parameter) and setting the value of $\alpha$ (elastic net mixing parameter) to zero. Finally, the naïve Poisson model with all predictors and no regularization was fitted to the data.

Model fit was assessed in the test data set ($N = 288$) using several descriptive statistics. First, residual variance was quantified using deviance residuals (*Friedman, Hastie & Tibshirani, 2010*) and mean squared error; second, the similarity of the predicted values and the actual values were assessed by calculating their Pearson correlation; third, McFadden's Pseudo R (*Mesken, Lajunen & Summala, 2002*; *Mittlböck & Schemper, 1996*) was used for assessing how much model fit improved from the null model; finally, the min–max index was calculated as

$$\text{MinMax} = \sum_i \left( \frac{\min(y_i, \hat{y}_i)}{\max(y_i, \hat{y}_i)} \right) \Big/ n.$$

## RESULTS

### Network analyses

The results reported below are based on the maximum number of cases available for the respective analyses as described in the Methods section ($N = 1{,}173$ for the between-person model and $N = 8{,}858$ for the cross-sectional network model, respectively). First, the between-person network is shown in Fig. 3. It illustrates between-person differences

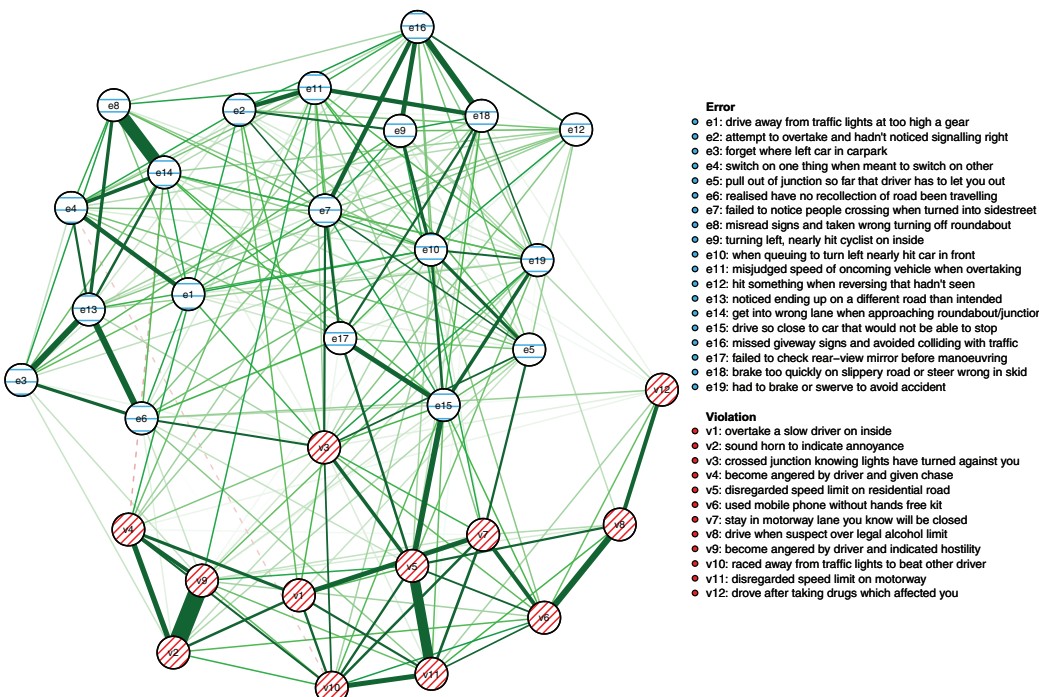

**Error**
- e1: drive away from traffic lights at too high a gear
- e2: attempt to overtake and hadn't noticed signalling right
- e3: forget where left car in carpark
- e4: switch on one thing when meant to switch on other
- e5: pull out of junction so far that driver has to let you out
- e6: realised have no recollection of road been travelling
- e7: failed to notice people crossing when turned into sidestreet
- e8: misread signs and taken wrong turning off roundabout
- e9: turning left, nearly hit cyclist on inside
- e10: when queuing to turn left nearly hit car in front
- e11: misjudged speed of oncoming vehicle when overtaking
- e12: hit something when reversing that hadn't seen
- e13: noticed ending up on a different road than intended
- e14: get into wrong lane when approaching roundabout/junction
- e15: drive so close to car that would not be able to stop
- e16: missed giveway signs and avoided colliding with traffic
- e17: failed to check rear–view mirror before manoeuvring
- e18: brake too quickly on slippery road or steer wrong in skid
- e19: had to brake or swerve to avoid accident

**Violation**
- v1: overtake a slow driver on inside
- v2: sound horn to indicate annoyance
- v3: crossed junction knowing lights have turned against you
- v4: become angered by driver and given chase
- v5: disregarded speed limit on residential road
- v6: used mobile phone without hands free kit
- v7: stay in motorway lane you know will be closed
- v8: drive when suspect over legal alcohol limit
- v9: become angered by driver and indicated hostility
- v10: raced away from traffic lights to beat other driver
- v11: disregarded speed limit on motorway
- v12: drove after taking drugs which affected you

**Figure 3** **The between-person network model.** The colors of the nodes correspond with the errors/violations dichotomy commonly used in DBQ studies. Green edges signify positive associations, dashed red edges negative associations. The wider and the more opaque the edge, the stronger the association.

in the connectivity of the driver behaviors during the first 3 years of learning to drive. Errors are shown as striped blue nodes and violations as striped red nodes. The presence of an edge between two nodes represents their conditional dependence when controlling for all other nodes.

First, in outline, violations and errors occupy different regions of the graph, and their distinction seems a sensible rough description of the data. However, if all violations and all errors were reflections of respective latent variables, we would expect all violations and all errors to be interconnected (*Epskamp & Fried, 2018*) and for all violations and errors to be independent. In contrast, thematically related violations are clustered together and connected by strong edges, aggression-related nodes (v2, v4, v9) and speeding-related nodes (v5, v11) being a case in point. The edges are interpreted as showing, for example, that drivers who were more likely than others to exceed speed limits within residential areas (v5) were also more likely to do so on highways (v11). Similarly, nodes related to substance abuse (v8, v12) were connected to each other and few other behaviors, except using the cell phone while driving (v6). Similar considerations apply to errors. For instance, the nodes related to forgetting something (e3, e6) shared a strong edge and were connected to few other nodes except absent-mindedness (v13).

Second, certain violations and errors were connected by relatively strong edges. Notably, drivers who exceeded speed limits within residential areas (v5) were more likely than others to tailgate other drivers (e15). Further, the tailgating drivers were more likely

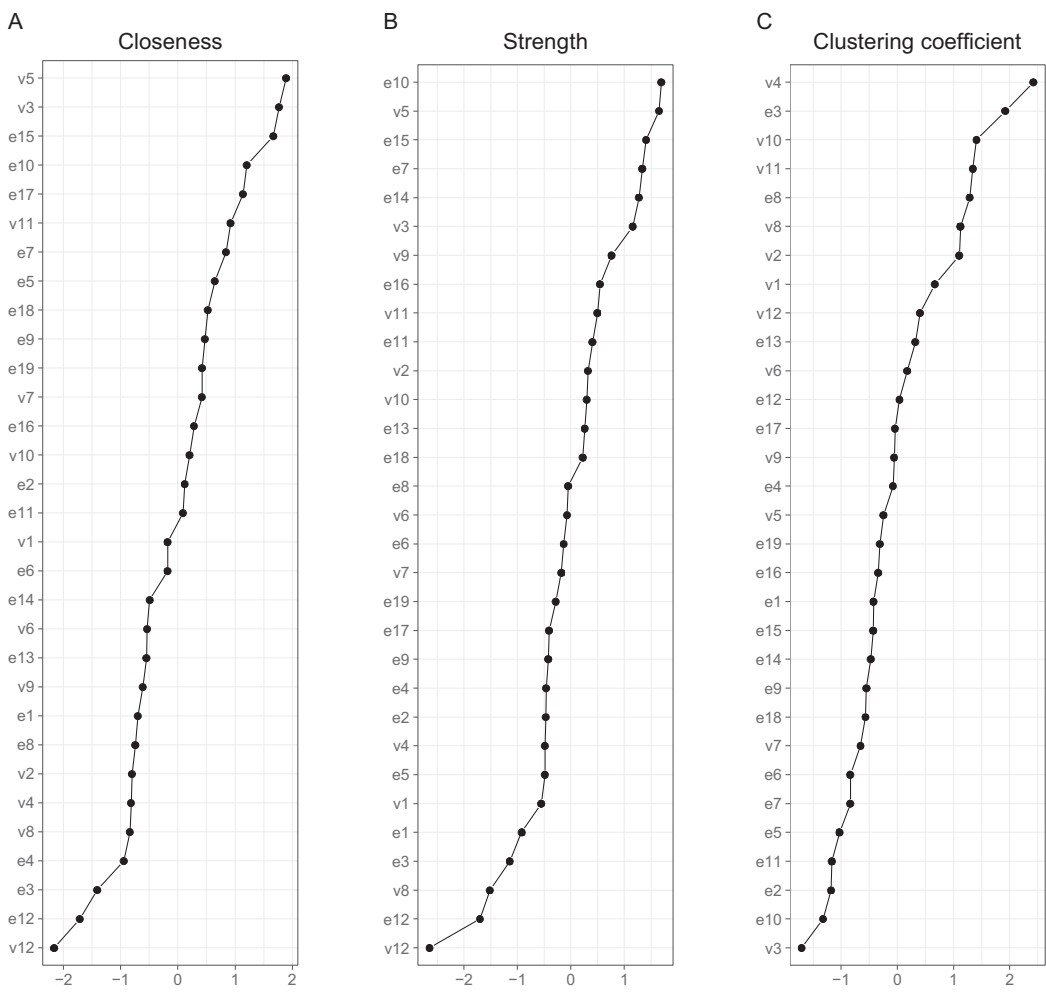

**Figure 4 Indices of centrality and clustering for the between-person network.** (A) Closeness centrality. (B) Strength centrality. (C) Zhang's local clustering coefficient.

to need to brake or swerve to avoid an accident (e19). Similarly, crossing junctions against a red light (v3, a violation) was connected by roughly equally strong edges to other violations and errors.

Third, not all thematically related nodes were connected by strong edges, the two items related to perceiving traffic signs (e8, e16) being a case in point. Still, both shared strong edges with other errors; for instance, misreading signs (e8) with getting into a wrong lane (e14). Finally, a methodological note is in order: differences between edge strengths can be interpreted only if their confidence intervals are not excessively wide. Bootstrap analyses assessing this are reported in Figs S4–S5).

Interpreting large networks such as those in Fig. 3 becomes easier with examining indices of node centrality and clustering. The interpretability of the indices themselves depends on their stability, which can be judged by calculating the CS coefficient based on bootstrap tests (Table S5). The indices whose CS-coefficient exceeded the recommended value of 0.5 (*Epskamp, Borsboom & Fried, 2017*) are shown in Fig. 4.

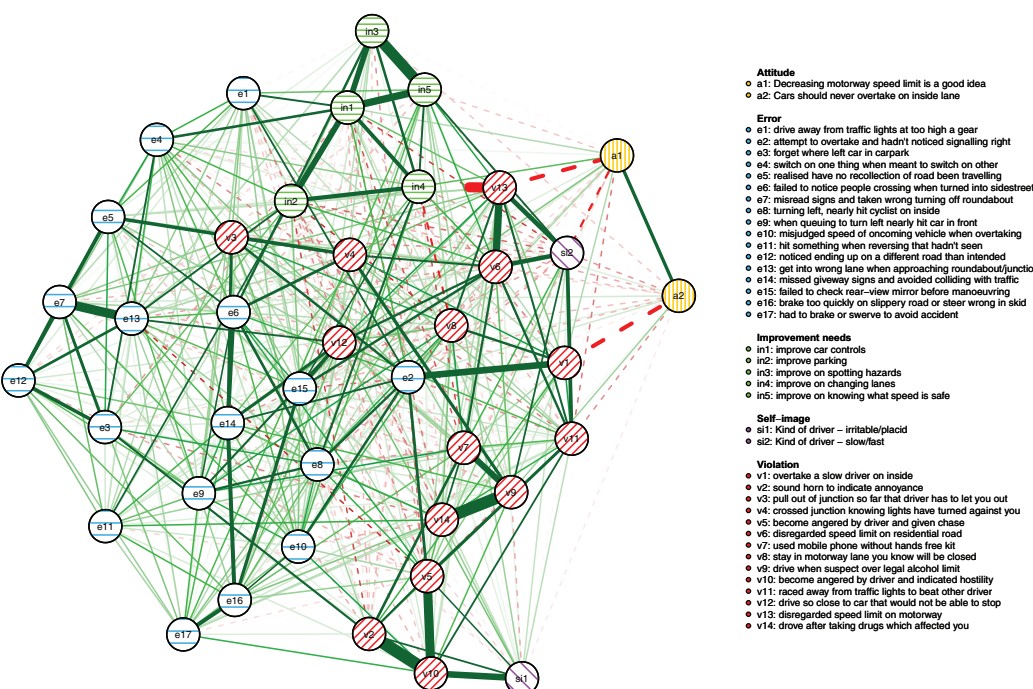

**Figure 5 The cross-sectional network model at the first time point (6 months post-licensure) showing the relationships of the DBQ variables and background variables.**

Both strength centrality (associations with immediate neighbors) and closeness centrality (associations with all nodes) of the between-person network indicated the presence of a group of nodes that were especially central in the network. Closeness centrality is perhaps the more revealing in this context, as it capitalizes less on single strong edges. The node with the highest closeness centrality was v5 (speeding within a residential area), followed by nodes v3 (crossing junction on red), e15 (tailgating), e10 (queuing, nearly hit car), e17 (fail to check mirror) and v11 (speeding on motorway); nodes e10, v5, e15 and v3 had also high strength centralities. In addition, nodes e14 (getting into a wrong lane) and e7 (failing to notice pedestrians) had a high strength centrality. In general, the nodes along the path connecting speeding with various errors (v11-v5-v3-e7-e17-e15-e10 or v11-v5-e15, etc.) were central. Figure 4 also shows Zhang's clustering coefficient, according to which certain nodes, most notably v4 (become angered, give chase) and node e3 (forget where left car) contained little unique information.

The cross-sectional network model (Fig. 5) shows the DBQ variables in relation to various background factors: attitudes (yellow), self-judged improvement needs (green) and self-image as a driver (lilac). The model is based on data collected at the first time point, 6 months post-licensure. It appears at the first sight quite different from the between-person model, but this is largely because the node placements are different due the use of the Fruchterman–Reingold algorithm. Similarities between the two models are revealed by examining the connection strengths and the centrality indices (Fig. 6). The indices whose CS-coefficient exceeded the recommended value of

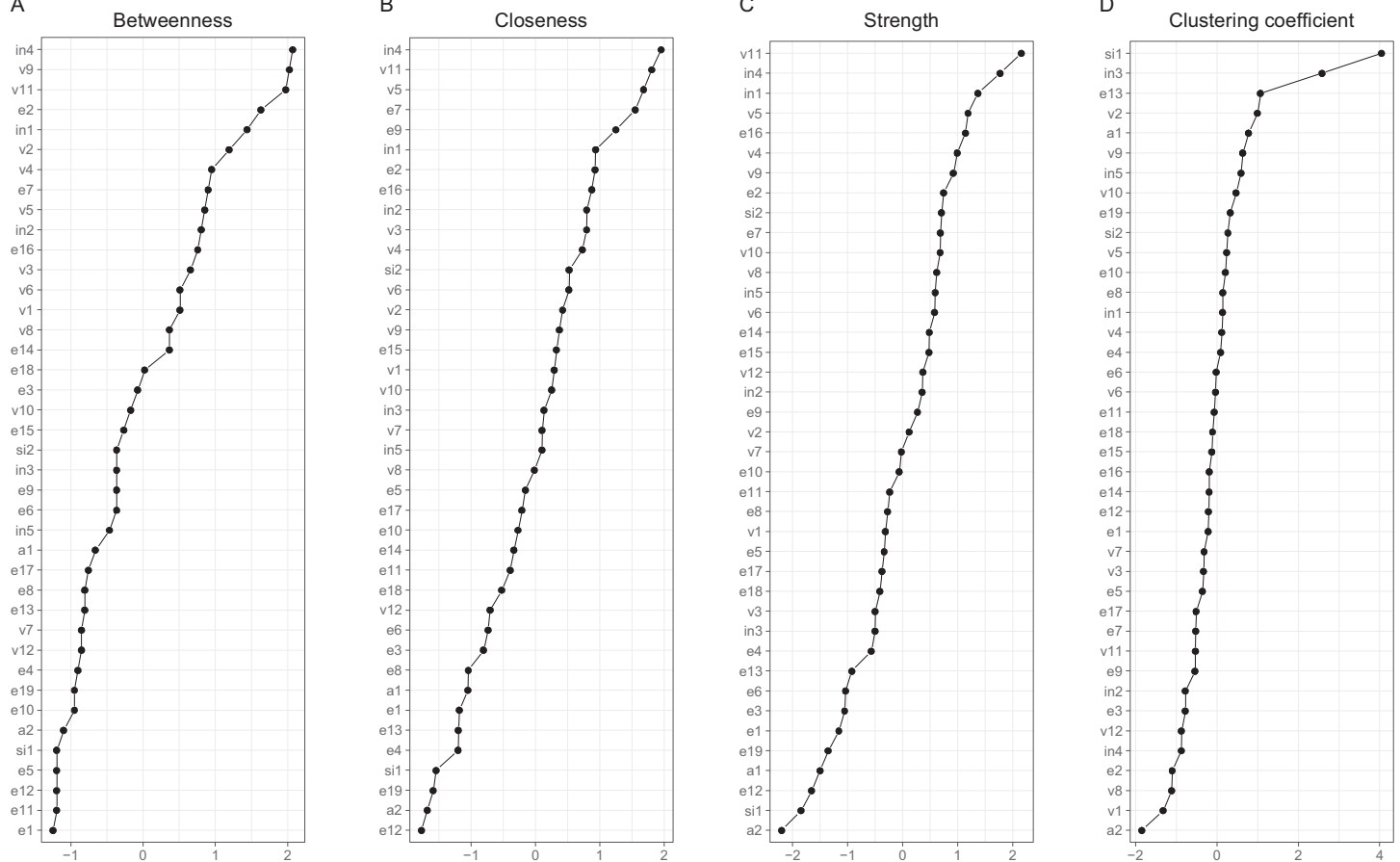

**Figure 6 Indices of centrality and clustering for the cross-sectional network.** (A) Betweenness centrality. (B) Closeness centrality. (C) Strength centrality. (D) Zhang's local clustering coefficient.           

0.5 (*Epskamp, Borsboom & Fried, 2017*) are shown in Fig. 6. The speeding-related nodes (v5 and v11) were again central together with node e7. Node e16 (missing give way signs) now had a high strength centrality, while in the between-person network the corresponding centrality value was average; on the other hand, the connectivity patterns of the node were similar. The path v11-v5-e15 was present also in this model, even though node e15 was now slightly less central. A prominent difference between the models was that the aggression-related nodes v2, v4 and v9 were more central in the cross-sectional model.

Looking at the background factors, the self-judged improvement needs were strongly interconnected. They were, however, also related to driving behaviors in a revealing pattern. In general, they shared negative associations with different violations. In particular, the judged need for improvement in changing lanes (in4) was related to less speeding (v11) and less pushing into a lane (v7), while the perceived need of improving controlling the car (in1) was associated with having problems with gears (e1) and car controls (e4). Negative attitudes to speeding (a1) and overtaking on the inside (a2) were related to fewer self-reported behaviors of those kinds (v11 and v1, respectively).

**Table 1 Regression model fit in the training and hold-out samples.**

| Variable | Elastic net | Ridge | Naive poisson |
|---|---|---|---|
| Training sample | | | |
| Pearson $r$ | 0.246 | 0.260 | 0.317 |
| Mean square error | 0.625 | 0.622 | 0.590 |
| Min–max index | 0.121 | 0.122 | 0.128 |
| McFadden pseudo $R^2$ | 0.030 | 0.033 | 0.057 |
| Deviance | 968.4 | 964.8 | 924.8 |
| Null model deviance | 1018.1 | | |
| Hold-out sample | | | |
| Pearson $r$ | 0.155 | 0.160 | 0.086 |
| Mean square error | 0.627 | 0.626 | 0.745 |
| Min–max index | 0.158 | 0.157 | 0.156 |
| McFadden pseudo $R^2$ | 0.010 | 0.011 | −0.044 |
| Deviance | 317.9 | 317.2 | 348.4 |
| Null model deviance | 323.6 | | |

On the other hand, perceiving oneself as a fast driver (si2) was positively associated with speeding (v5 and v11) and racing from the lights (v10). The drivers' self-judgment of themselves as irritable (si1) was closely associated with the anger-related nodes, even though the node was quite redundant as judged by the clustering coefficient. Self-perceived need of improvement in hazard-perception (in3) had a similarly high value of the clustering coefficient.

### Regression analysis

The self-reported driving behaviors, age, sex and mileage at the first time point were used in predicting subsequent crashes. As seen in Table 1, the naive Poisson model fit the training data best (as it should), but its performance deteriorated notably in the hold-out data. In fact, the model fit the hold-out data worse than the null model without predictors, offering a dramatic demonstration of how such models sometimes overfit data. The elastic net model fit the training data the worst, but its fit to the hold-out data was almost identical to that of the ridge regression model and is to be preferred due to its parsimony (16 vs. 42 parameters, respectively).

Table 2 shows the regression weights of the elastic net model that are low, partly because of the dual penalization involved. The driving behaviors included in the model comprise various violations and errors. Variables related to failing to notice pedestrians, problems with car controls and different forms of aggressive behavior had the highest regression weights. Hitting something when reversing, which is itself a minor crash, also predicted subsequent crashes.

### DISCUSSION

The present paper put forward three main ideas: (1) Individual violations and errors correlate because direct, possibly causal, relationships exist between perceptions, thoughts, emotions and actions in traffic. In particular, individual violations and errors

**Table 2 Regression weights of predicting accidents based on the elastic net regression analysis.**

| Variable | Regression weight |
|---|---|
| Mileage | 0.04 |
| Gender | |
| Age | |
| Drive away from traffic lights at too high a gear | 0.07 |
| Overtake a slow driver on inside | 0.04 |
| Have to confirm you're in right gear | |
| Attempt to overtake and hadn't noticed signalling right | |
| Forget where left car in carpark | |
| Sound horn to indicate annoyance | 0.07 |
| Switch on one thing when meant to switch on other | |
| Change into wrong gear when driving along | |
| Pull out of junction so far that driver has to let you out | 0.04 |
| Have used plates to warn drivers you are a new driver | |
| Realized have no recollection of road been travelling | |
| Crossed junction knowing lights have turned against you | 0.01 |
| Failed to notice people crossing when turned into side street | 0.08 |
| Become angered by driver and given chase | |
| Misread signs and taken wrong turning off roundabout | |
| Drive in either too low or high gear for conditions | |
| Disregard speed limit on residential road | 0.04 |
| When turning left have nearly hit cyclist on inside | |
| Used mobile phone without hands free kit | |
| Stay in motorway lane know will be closed | |
| When queueing to turn left nearly hit car in front | |
| Drive when suspect over legal alcohol limit | |
| Forget to take handbrake off before moving off | 0.03 |
| Become angered by driver and indicate hostility | 0.06 |
| Misjudge speed of oncoming vehicle when overtaking | |
| Hit something when reversing that hadn't seen | 0.06 |
| Raced away from traffic lights to beat other driver | 0.01 |
| Used a hands free kit | |
| Selected wrong gear when wanting to go in reverse | |
| Noticed on different road to destination want to go | −0.01 |
| Get into wrong lane when approaching roundabout/junction | |
| Drive so close to car that wouldn't be able to stop | |
| Forget headlights were on full beam | |
| Missed give-way signs and barely avoided collision | 0.02 |
| Disregarded speed limit on motorway | 0.01 |
| Failed to check rear-view mirror before manoeuvring | |
| Brake too quickly on slippery road or steer wrong in skid | |
| Drove after taking drugs which think affected you | |
| Brake/swerve to avoid accident | 0.01 |

were not assumed to reflect the effects of underlying, in principle unobservable variables, such as violation-proneness and error-proneness. (2) Hypotheses concerning these relationships can be formulated in a data-driven manner by estimating the strengths of associations between pairs of variables while controlling for all other variables in a network model. Similar ideas of network models as hypothesis-generating structures have been expressed in the context of psychopathology research (*Fried & Cramer, 2017*; *Borsboom, 2017*). (3) Background factors such as attitudes, self-image, etc. are directly related to individual traffic behaviors such as speeding or drunk driving; similarly, individual traffic behaviors are related to crash risk, and these relationships can be explored by developing criterion-keyed psychometric scales (*Chapman, Weiss & Duberstein, 2016*). Finally, to understand different errors and violations, one needs to consider the context in which they take place rather than assuming that they reflect stable traits of the individuals (error-proneness and violation-proneness). The empirical part of the study dealt with the first three questions. Research related to the fourth is also discussed, as the context-sensitivity of behavior is an important motivation for network models of human behavior.

The dynamics of the relationships between the typical levels of individual violations and errors were examined by constructing the between-subject network. The analysis showed that drivers who were more likely than others to exceed speed limits within residential areas were also more likely to tailgate the vehicle in front; further, the tailgating drivers were more likely than others to need to brake or swerve to avoid an accident. These associations were interpreted as causal hypotheses: drivers who exceed speed limits will likely catch up with the traffic flow and may as a consequence need to react. Similarly, crossing a junction against a red light (a violation) was associated with failing to notice pedestrians and with other errors. Importantly, such associations between violations and errors are difficult to accommodate within latent variable models of driver behavior.

Exceeding speed limits within residential areas appeared as the most closeness central and the second-most strength central node in the between-subject network. Insofar as the centrality of a node can be taken to reflect the causal connections emanating from that node, a successful intervention of reducing speeding might affect the other behaviors directly or indirectly linked to it. Care must be taken with such interpretations, since the edges in the network may well reflect other factors than a cause-effect relationship (*Epskamp & Fried, 2018*; *Kossakowski et al., 2016*; *Golino & Epskamp, 2017*; *Epskamp, Kruis & Marsman, 2017*; *Epskamp et al., 2018*), but at the very least, such causal hypotheses can be formulated based on the present results. In fact, network models benefit traffic research precisely in that they encourage thinking of the dynamics of the behaviors. In contrast, latent variable models remain silent in this respect, as they conceptualize individual behaviors as causally passive indicators of the latent variables (*Borsboom, Mellenbergh & Van Heerden, 2003*). Thus, if we take the latent variable view seriously, we can only influence individual behaviors through manipulating the latent variables: whether we want to reduce drunk driving or speeding, we should aim at the drivers' rule-breaking tendencies, because influencing an individual violation has no effect on other behaviors under the latent variable view.

The relationships between background factors and traffic behaviors were examined in the cross-sectional network model based on data collected 6 months post-licensure. Drivers who perceived themselves in need of improving lane-changing skills were less likely to report speeding and pushing into a lane. Further, the respondents reported behaving according to their attitudes: a preference of decreasing speed limits was associated with less speeding. On the other hand, perceiving oneself as a "fast driver" was associated with more speeding and racing from the lights. In short, individual background factors were related to driver behaviors in an understandable pattern.

Even though it is difficult to explain direct associations between driver behaviors using latent variable models, they pose a challenge to the network view: certain behaviors are more likely to occur together than others; why is this so if there are no latent variables? One can begin to answer by considering relationships between behavior and environment, as illustrated by the following quote from a study on driver irritation and aggression:

"Drivers who enjoy a somewhat faster speed than other drivers will more often be obstructed by other traffic, and therefore they will become irritated more often and be more likely to educate other road users. They probably also will become more irritated than other drivers when obstructed, because they want a faster progress"

(*Björklund, 2008*).

In other words, people have characteristics such as emotions, attitudes and personality components (*Cramer et al., 2012a*) that affect their behavior in traffic, which is not only something to be explained, but also a variable that feeds back into the system of emotions, perceptions and other behaviors. Further, not all behavioral characteristics are equally compatible with each other. For instance, the people described in the quote may be unlikely to make errors related to car controls, which are perhaps related to lack of experience or lack of interest in cars. In technical terms, network models exhibit non-trivial topology (*Borsboom, 2017*). Further, they can—as also demonstrated in the present study—accommodate background variables such as beliefs, and account for why beliefs, feelings and behaviors become aligned (*Dalege et al., 2016*). For instance, the drivers described in the quote are perhaps likely to consider speed limits in general too low to avoid cognitive dissonance between their actions and beliefs.

The above example illustrated stable differences between people. However, people do not always behave in a stable manner; rather, their behavior is context-dependent. The idea has much in common with the cognitive-affective personality system (CAPS) (*Mischel & Shoda, 1998*) theory developed within personality psychology, which posits if-then rules that describe how someone typically reacts in a certain type of a situation. CAPS is influenced by connectionist models, and characterizes behaviors, memories and emotions as differently activated by the situational context and each other. The interconnected elements are described as a network in which activation is sustained by feedback loops. Such situation-specific rules might apply to, for example, young people driving with their friends vs. with a mother and a baby on board. In the former situation, the driver's repertoire of certain risky behaviors is more highly activated and activation spreads through excitatory links. For example, the threshold of speeding

might be lowered, activating an impulse to race other drivers. In short, the network of driving behaviors can be seen as being in two qualitatively different states distinguished by the connection strengths between the behaviors. As another example, it has been shown that young mothers have an elevated crash risk when driving with an infant passenger compared to driving alone (*Maasalo, Lehtonen & Summala, 2017*); the dynamics of their behavior in these contexts are likely to differ.

In addition to presenting network models of driver behavior, this study involved predicting crashes from individual behaviors. In contrast, previous self-report studies have mainly used latent variables in crash prediction (even though see *Warner et al., 2011* for an exception). The novel contributions of the present study were threefold. First, crashes were truly predicted from earlier behavior. Second, predictive models were first fit in a training subsample and then verified in an independent subsample of data. Third, regularized regression was used to prevent overfitting.

Three predictive models were tested, out of which the elastic net model (*Zou & Hastie, 2005*) and the ridge regression model fit the hold-out data roughly equally well, and notably better than the naïve Poisson regression model. The elastic net model has the benefit of being more parsimonious than the ridge regression model, so it is to be preferred, other things equal. In the elastic net model, predictors included displaying anger while driving and driving fast (disregarding speed limits and racing from traffic lights). Further, errors in visual search (failing to notice people, missing signs) and in controlling the car (driving off at wrong gear, forgetting the handbrake) were included in the model. Interestingly, hitting something when reversing, which is itself a minor crash (*De Winter, Dodou & Stanton, 2015*), predicted future crashes. Remarkably, the naïve Poisson model including all predictors fit the hold-out subsample worse than the null model with no predictors, offering a dramatic demonstration of the dangers of overfitting.

The present study has its limitations. First, network models are motivated by modelling the *components* of a phenomenon, with a component defined as something having unique causal relations with the rest of the network (*Cramer et al., 2012a*). The DBQ has been psychometrically developed to maximize reliability, which has resulted in a certain redundancy of the items. In this study, this shows as high values of the clustering coefficient for the nodes related to aggression and speeding, which could in future studies be represented by single nodes. On the other hand, developing a novel questionnaire describing potentially causally related behaviors, thoughts and emotional reactions is another option. Further, the behaviors examined here are influenced by other road users, which could not be accounted for. For instance, aggressive behavior is difficult to understand without knowing something about its target. It is naturally possible to take an even more critical perspective and argue that none of the relationships indicate potential causal relationships and driver behaviors should continue to be viewed as being caused by latent variables. Even so, the present study presents a challenge: why are variables that commonly load on different factors strongly correlated? Finally, it is likely that general psychological characteristics, such as impulsivity, conscientiousness, neuroticism, agreeableness, attention and memory capacity, etc. would explain the

associations observed in the present study. The fact that information on such characteristics was not available is a clear limitation of the present study.

In addition to the substantive questions, some methodological decisions were problematic. First, polychoric correlations that estimate normally distributed variables underlying ordinal input variables were used, even though the variables were positively skewed. Although the practice is widespread in psychometric network models, it is not optimal (*Epskamp, 2017*). Further, listwise deletion of missing data was performed; less wasteful methods are under development for network models but not yet available (*Epskamp, 2017*). Further, it has been shown that young respondents may give exaggerated responses to questions they find funny, a response bias dubbed "mischievous responding" (*Robinson-Cimpian, 2014*); this could affect items such as drug use while driving. On the other hand, if such biases are transient in nature, between-person networks are likely a suitable method to use (*Costantini et al., 2019*).

Choosing the correct components of driver behavior is a central issue to be tackled in future network models of driver behavior. In addition, future studies should aim at developing a network theory of driver behavior instead of merely applying a novel modelling tool. A recently developed intricate error taxonomy (*Stanton & Salmon, 2009*) might provide a good starting point together with factors contributing to such errors. Also, the need for self-report studies will remain in the future even though studies involving instrumented vehicles have become ever more intricate (*Precht, Keinath & Krems, 2017*), because physical measurements are underdetermined psychologically (*Transport Research Laboratory, 2015*): for instance, a sudden acceleration can be due to either racing from the lights or unfamiliarity with car controls.

Further, the context-dependent nature of driving needs to be taken into account in future studies. Drivers may behave differently depending on who they are traveling with; similarly, investigating drivers' developing situation-awareness of when to desist from violating rules has been called for (*Transport Research Laboratory, 2015*), and network models are an ideal tool for investigating such developmental trajectories (*Epskamp et al., 2018*). Another fascinating future direction is to consider the effects of the driver's state: being fatigued, intoxicated, stressed, in a hurry or in a strong emotional state could conceivably cause the network of driver behaviors to occupy qualitatively different states. Existing task analyses and models of driving situations (*Fastenmeier & Gstalter, 2007*; *Oppenheim & Shinar, 2012*) are likely to be a good starting point, because individuals may behave in a more-or-less stable manner in a certain type of a situation, but not across situations (*Mischel & Shoda, 1998*). Intriguing avenues for future research await those willing to look into the networks of driver behavior.

## CONCLUSIONS

Representing violations and errors on the road as interconnected networks of behaviors, cognitions and emotions makes it possible to formulate data-driven hypotheses on causal connections between individual violations and errors. For instance, exceeding speed limits may make it more likely for drivers to end up tailgating other vehicles, which in turn may make it more likely that they need to brake or swerve to avoid accidents.

In contrast, previous psychometric work has been based on the latent variable view, according to which individual errors and violations function as (nothing but) reflections of underlying psychological properties, error-proneness and violation-proneness. It is argued herein that this is an overly simplified view of the determinants of traffic behavior, and that the network view provides a useful novel point of view in this respect.

More generally, network models show promise for bridging a gap between experimental and theoretical work in traffic research on the one hand and self-report-based research on the other hand. The latter has commonly assumed the existence of a small number of mutually exclusive psychological mechanisms whose operation is reflected in respective sets of driver behaviors (e.g., violations and errors) that can be represented using latent variables or sum scores. On the other hand, the importance of individual driver behaviors, such as speeding, is recognized in theories of driver motivation (*Fuller, 2005*; *Summala, 2007*), studies that aim at determining reasons for speeding (*Lawton et al., 1997*; *Parker et al., 1992*; *Warner & Åberg, 2006*) and engineering models of accidents (*Abdel-Aty & Radwan, 2000*). Similarly, an error taxonomy involving action errors, cognitive and decision-making errors, observation errors and information retrieval errors has been proposed (*Stanton & Salmon, 2009*), indicating the need to differentiate errors in a fine manner. Network models that focus on individual driving behaviors and their interrelationships offer a novel point of view from which to integrate the results of self-report studies with these lines of research.

In addition to presenting network models of driver behavior, this paper involved predicting crashes based on individual errors and violations. This was done using cross-validated penalized regression analysis, which resulted in a model that was both predictive of accidents and generalizable to new data. Similarly to the network models, the predictive models can be contrasted to those used in prior psychometric traffic research, in which latent variables have been used as predictors of crashes: the current paper argues that it is important to consider the role of individual traffic behaviors in determining crashes and to build predictive models from this point of departure.

## ACKNOWLEDGEMENTS

I would like to thank Jami Pekkanen and Otto Lappi for constructive feedback on earlier versions of the manuscript and Jami Pekkanen for helpful discussions on performing the regression analyses.

### Funding

The author received no funding for this work.

### Competing Interests

The author declares that he has no competing interests.

# PeerJ

## Author Contributions

- Markus T. Mattsson conceived and designed the experiments, performed the experiments, analyzed the data, contributed reagents/materials/analysis tools, prepared figures and/or tables, authored or reviewed drafts of the paper, approved the final draft.

## Human Ethics

The following information was supplied relating to ethical approvals (i.e., approving body and any reference numbers):

The study is based on archival questionnaire data collected in the United Kingdom. Informed consent was inferred from returned postal questionnaires in accordance with the social research guidelines of Department of Transport.

In Finland, ethical review is not required for studies that are based on public documents, registries or archival data. The present study is based on archival data, and no ethical review was thus requested. The ethical principles can be accessed on the web page of the Finnish national board on research integrity (www.tenk.fi/en) at http://www.tenk.fi/sites/tenk.fi/files/ethicalprinciples.pdf.

## Data Availability

The raw data for the present work is available at the UK Data Service repository: https://discover.ukdataservice.ac.uk/doi?sn=5985#2.

The analysis codes used in the present contribution is available at https://osf.io/4wbmt/.

## Supplemental Information

Supplemental information for this article can be found online at http://dx.doi.org/10.7717/peerj.6119#supplemental-information.

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
