# Peer review of "Network models of driver behavior"

_PeerJ, doi:10.7717/peerj.6119_

## Round 0.1 · original submission · Major Revisions

Please answer all comments raised by the reviewers.

Reviewer 1 ·

Basic reporting

The article is well-written, with a clear structure and relevant citations. The scientific background is extensive, with sound theoretical considerations on why the network model can be interesting within the context of driver behavior.
The raw data are available and the figures are clear. However, the author could expand some more on how he came up with the relationships within the hypothetical network model of traffic behavior from figure 2 (are they based on anything else than the relationships in figure 1?).

The paragraph from line 77 can be a bit confusing at some points when looking at figure 1. Some of the (problematic) theoretical assumptions and conclusions explained in that paragraph seem more related to one-factor common cause models than to latent variable models in general. I find that these problems from a latent variable perspective and the problems from a one-factor common cause model are sometimes mixed up within the manuscript. The latent variable view of violations and errors from figure 1 already seems more expanded than the one-factor common cause models being used in latent variable models for psychopathology or intelligence, where the assumption is that all observable scores (from an IQ test or all depression symptoms) are caused by only one latent variable. Therefore, I don’t fully agree with the author that all assumptions described in this paragraph hold for the model of figure 1– for example the assumption that the variation in the observed items can only be explained by variation in the latent variables ‘violations’ or ‘errors’, or measurement error, since the latent variables themselves are influenced by some other higher-order (latent) variables (called “background factors”).
However, there are more specific problems related to latent variable models which do relate to the model in figure 1 and are very well described, such as the problem that the model of figure 1 doesn’t allow direct influences between the observable variables and gives no direction for interventions. Personally, I would suggest the author to describe the more specific problematic assumptions related to the latent variable view of violations and errors (figure 1) and to use the fact that this model already is quite an extensive model from a latent-variable perspective as a way to enucleate the urge for a model that accepts direct interactions, such as network models. This is a minor revision to clarify the text without altering the scientific relevance of this study or the conclusions.

Experimental design

The experimental design of the analyses is very well described, making it understandable for the reader what the author has done and with what intentions. The methods are described with sufficient detail. The study seems easily replicable, with freely available data and clear and comprehensible R-code. The most important statistical results are and well interpreted.
One minor revision would be to better explain the difference between the ‘cross-sectional network’ and the ‘between-person network’. Although the author is right in the difference between these networks and correctly citing the paper by Epskamp, Waldorp, Mottus & Borsboom (2018) where these distinct networks are introduced, they are introduced within the context of time series networks. Without more explanation, the name of these two types of networks are confusing, since normally between-person networks are based on cross-sectional data.

Validity of the findings

The conclusions are well stated and linked to the original research question. The discussion is extensive with good suggestions for future research. I agree with the call of the author for future research regarding forming a theoretical network model of driver behavior, however, applying the network model to empirical data of driving behavior is a very nice first step for the field.
One suggestion for the network comparison is to include a figure where the networks are plotted in the same lay-out, to better see the differences between them.

Additional comments

This is a well-written manuscript, with an original application of network theory. I recommend this manuscript for publication with minor revisions.

·

Basic reporting

1) The manuscript could benefit from a thorough double-check of its structure. For example, information about analyses is sometimes scattered throughout the methods and results section. In particular, the current structure of the “Measures” (L. 158-175) section could be improved, because it includes information that should be given in the “Statistical Analyses” section (e.g., which items were used in which analysis) and information that should be better given in a “Procedure” section (e.g., the fact that the “Learning to drive questionnaire” was administered before the DEQ).
In the materials section, please simply provide a list of all instruments and, for each instrument, a list of all constructs assessed by it, with an example of item for each.

Experimental design

2) L. 158-175. Why using different subsets of items in different analyses? For example, only a subset of 31 DBQ items was included the network analysis, whereas more items were included in the regressions. The choice of omitting an item from an analysis is very important and should be justified accordingly.

3) L. 152-157. I could not understand the exclusion criteria. “Cases with non-missing data in all variables were included, resulting in a sample size of 8858.” How did these 8858 respondents become 1173 and then 1152, if they did had no missing data in all variables? Furthermore, since the differences between the two final sample sizes (N = 1173 vs. 1152) are very small, it might be useful to use the same sample of 1152 for all analyses. This would guarantee that all differences observed between the two networks are due to the introduction of extra variables, and not to the slightly different samples considered.

4) L. 129-130. The between-person network does not represent “differences between drivers”, but the dynamics involving the typical levels of a trait (Costantini et al., 2017) (see also L. 257-258). The authors present a between-person network computed by averaging data across four timepoints and a cross-sectional network computed by using only the first timepoint. However, the cross-sectional network is also a between-subject network which however uses only one timepoint instead of four to estimate the DEQ variables. My suggestion would be to consider a single between-subject network in which DEQ variables are estimated as the average across the four timepoints, whereas the variables assessed only once (e.g., sex, attitude etc.) are included in the network as they are.

5) Why performing tuning parameter selection using EBIC for network analysis and Cross-validation for regression analysis?

6) Given the longitudinal nature of the data, why not considering also a within-subject network (Costantini et al., 2017), informing on the relationships among violations/errors within occasions, after controlling for each driver's typical beahvior? This would be more in line with what is stated in the discussion section (e.g., L 438-453)

7) L. 179-181. Why choosing a subset of background factors, when the network estimation method includes a lasso regularization that allows performing predictor selection automatically?

Validity of the findings

no comment

Additional comments

There are many strengths in this manuscript, such as the use of state of the art methods such as network models and regression with tuning-parameter selection, to model a socially-relevant phenomenon such as driving behavior. Furthermore, I agree with the authors that the network approach is particularly suited to model this type of behaviors, which are arguably causally interconnected. In sum I found the manuscript very interesting and I think that it is should be published.
However, I also think that the manuscript may benefit form a careful revision in its structure (as I detailed in the Basic Reporting section). Furthermore, several methodological choices need a stronger justification (as i detailed in the Experimental design section). I also have a few general comments, that I present here in order of importance:


8) I think that a limitation of this study is that psychological variables other than those directly related to driving were not considered. General psychological characteristics (e.g., impulsiveness and low conscientiousness, attention, memory) might explain part of the relationships identified in this study. A potential future development of this study could involve assessing these variables and inspecting their role within the network. I expect that some of these variables might explain some of the relationships emerged in this study. Notice that this would not be the same as hypothesizing abstract "latent" variables based on driving behavior alone (e.g., violation proneness) and that it would be therefore compatible with a network approach, both theoretically and methodologically.

9) L 61-116. The latent-variable models are represented in a simplistic way and perhaps overly criticized in the introduction. For example, albeit Reason et al. (1990) have considered only a model with two latent variables, one could have opted for a more complex Structural Equation Model (SEM) including a larger number of latent variables and path relations among them, to represent complex relationships. Furthermore, the causal interpretation of latent variables is not the only possible one, e.g., they can be considered statistical summaries that are useful for several practical purposes. When presenting latent variable models, it could be useful to consider also some of their strengths (e.g., they provide simple models and simple assessment procedures). In sum, I think that the point should not be that latent variable models are flawed, but that network models can provide additional information. Although network models allow representing relationships that cannot be well modeled by SEM alone, this does not mean that latent variable models cannot be also useful (e.g., see Epskamp, Rhemtulla, & Borsboom, 2017).

10) L. 118-125. Other possible interpretations of a nonzero edge include the presence of an unmeasured (latent) variable affecting both nodes (Epskamp, Kruis, & Marsman, 2017) and the inclusion of a common “collider” (i.e., a variable that is caused by two variables) in the model (Epskamp, Waldorp, Mõttus, & Borsboom, 2018; Lee, 2012).

11) L. 236-243. Why not considering also a lasso regression model, consistent with network estimation?

12) L. 256-257. What it is meant by “The results reported below are based on the maximum number of cases available for the respective analyses”?

13) L. 388-389. Please, specify that the node is the most central only according to closeness. It would be useful to specify that the success of the intervention would depend on the assumption that the centrality of the node reflects causal connections emanating from that node.

Minor comments:
- L. 34. Please, provide reference for the DBQ when first mentioned.
- L 43-49. The three quotations by Reson et al. (1990) are somehow redundant. Please, summarize them in a single sentence.
-L. 140-150. Please clarify the longitudinal nature of data, I understood that the data were not cross-sectional only when reading L. 182.
- L. 237. For consistency, use GCVE instead of “deviance statistic”.
- L. 244-247. It would be useful to specify that model fit was computed on the validation data (N = 288).
- L. 236-243. The letter lambda was used to indicate the lasso parameter for networks and the ridge regression parameter for the regression analysis, whereas in regression analysis alpha is used to indicate the lasso parameter. Would it be possible to use a different notation (e.g., using lambda always to indicate lasso penalty) to avoid confusion?
- L. 348. Please, specify “parsimony” better. How many parameters did each model include?
- Figure 1. Why including predictors only for Violations in the example?
- Figure 2. Why including congestion and stress, which are not in Figure 1?

---

## Round 0.2 · accepted · Accept

All comments have been addressed.

# Reviewer 1 ·

Basic reporting

The author made thorough revisions on the introduction, especially regarding the following three topics: Difference between measurement model and structural model, difference between general latent variable model and one-factor latent variable model and added a well-described paragraph about ontology and the consequences of a realist interpretation of factor models. With this revision, my former concerns regarding these topics have been met.

Experimental design

My concerns regarding the discussion by the author of between-subjects networks and cross-sectional networks have been met by the author, by clarifying that the data are longitudinal in nature and discuss how the networks are related.

Validity of the findings

The author replied with good arguments on why my suggestion regarding plotting the two networks in the same layout has not been incorporated, therefore I accept the choices made by the author.

Additional comments

The author replied extensively and satisfactorily to my raised concerns.

·

Basic reporting

no comment

Experimental design

no comment

Validity of the findings

no comment

Additional comments

I think that the manuscript, which I found very interesting in its first version, has been greatly improved and is now suitable for publication.
I only have a (very) minor comment: L 52-57. How is the possibility that “individual violations and errors may be causally related” a reason for which the latent variable perspective “has gone unchallenged”?